# Exploiting Asymmetry for Synthetic Training Data Generation: SynthIE and the Case of Information Extraction

**Martin Josifoski, Marija Šakota, Maxime Peyrard, Robert West**
EPFL
{martin.josifoski, marija.sakota, maxime.peyrard, robert.west}@epfl.ch

## Abstract

Large language models (LLMs) have great potential for synthetic data generation. This work shows that useful data can be synthetically generated even for tasks that cannot be solved directly by LLMs: for problems with structured outputs, it is possible to prompt an LLM to perform the task in the reverse direction, by generating plausible input text for a target output structure. Leveraging this asymmetry in task difficulty makes it possible to produce large-scale, high-quality data for complex tasks. We demonstrate the effectiveness of this approach on closed information extraction, where collecting ground-truth data is challenging, and no satisfactory dataset exists to date. We synthetically generate a dataset of 1.8M data points, establish its superior quality compared to existing datasets in a human evaluation, and use it to finetune small models (220M and 770M parameters), termed *SynthIE,* that outperform the prior state of the art (with equal model size) by a substantial margin of 57 absolute points in micro-F1 and 79 points in macro-F1. Code, data, and models are available at https://github.com/epfl-dlab/SynthIE.

## 1 Introduction

Large language models (LLMs) have demonstrated the ability to generate highly fluent and coherent textual data. One promising application of this capability is the generation of large amounts of high-quality synthetic data for training and evaluating smaller models. This becomes particularly useful for tasks where high-quality datasets are not readily available or access to real data is limited or expensive. However, in many complex natural language processing (NLP) tasks, the textual input $x$ is mapped to a *structured* (rather than free-text) output $y$, and in such cases, LLMs may perform poorly as synthetic-data generators, since pretraining did not gear them to produce the specific required output format (even with in-context learning). Here

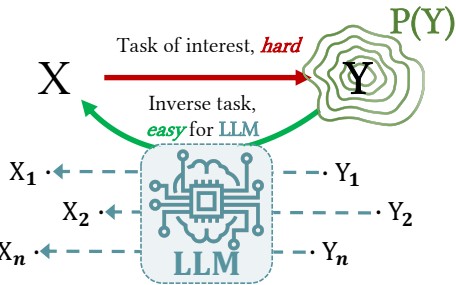

Figure 1: **Exploiting asymmetry for SDG.** For hard tasks of interest with input $X$ and output $Y$, the reverse task (from $Y$ to $X$) may be much easier for an LLM. If so, we can generate high-quality training pairs $(X,Y)$ by prompting an LLM to generate plausible inputs $X$ from outputs $Y$. This often holds true for tasks with structured $Y$, as in closed information extraction, where $X$ would be the input text and $Y$ would be the list of (subject, relation, object) triplets expressed in the input text. Furthermore, this ensures full control over the sampling distribution $P(Y)$, and thus balanced datasets.

we propose to alleviate this issue by generating synthetic data in the reverse direction by first sampling an output structure $y$ and then prompting the LLM to generate a corresponding input text $x$ (see Fig. 1). We postulate that for many tasks, given appropriate in-context information, an LLM can generate a meaningful $x$ corresponding to a structure $y$, even when the original task cannot be solved directly by the LLM. Exploiting this asymmetry, then, will allow us to synthetically generate high-quality data even for hard tasks. Furthermore, the flexibility to choose the distribution over output structures $y$ gives us fine-grained control over the data.

A good example of such a task, on which we focus in this work, is *closed information extraction* (cIE). In cIE, a model must extract a set of disambiguated triplets (i.e., facts) $y$ from a given text $x$. These triplets are of the format (subject, relation, object), with the subject and object being entities in

a knowledge base (e.g., Wikidata) and the relation being specified in the knowledge base schema.

Collecting datasets for this task is time-consuming and expensive, as it requires annotators to know the entire entity and relation catalogs and reason about all possible facts expressed in the text $x$. As a result, only small or noisy datasets exist to date. The largest dataset available, REBEL (Huguet Cabot and Navigli, 2021), suffers from several problems (Josifoski et al., 2022): (i) Noise: it is collected with a mix of heuristics, and for many data points, the target output $y$ does not contain all the facts expressed in the input $x$ or is (partially) incorrect. (ii) Skewness: most relations appear only rarely in the dataset, which results in models that ignore most of the information when the data is used for training and in poor performance estimates when it is used for evaluation.

Importantly, cIE is also not solvable directly by LLMs such as GPT3.5, as such models are unaware of the entity and relation identifiers of interest (for examples of failures, see Appendix A). We show that, although the LLM cannot be used to produce output structures directly, we can use it to generate high-quality input texts by reversing the task.

**Contributions.** (i) We propose a *strategy* for designing effective *synthetic data generation (SDG) pipelines* and apply it to cIE. Concretely, we start by sampling triplet sets from the Wikidata knowledge graph (KG) such that each triplet set is coherent (i.e., can be expressed by a short text) and the selected triplet sets cover all relations more uniformly than REBEL does. For each triplet set $y$, we then prompt an LLM to generate text $x$ that expresses these, and only these, triplets. This results in a high-quality synthetic dataset (1.8M data points) that we use to replace the noisy REBEL dataset. The process is illustrated in Fig. 2.

(ii) In a human evaluation, we show that our synthetically generated data is of significantly *higher quality* than the existing REBEL dataset, has a *more uniform relation-frequency distribution,* and can be *generated cheaply* at scale. The evaluation reveals that, with 70% of the information from the text not present in the target set and 45% of the target triplets not actually expressed in the text, REBEL's test set cannot be used for an accurate estimation of performance. In contrast, for our highest-quality test set, the corresponding numbers are 15% and 22%, respectively.

(iii) We introduce *SynthIE,* a series of Flan-T5 models (Chung et al., 2022) finetuned on our synthetic data. In contrast to previous models, which perform well only for the few most frequently occurring relations, SynthIE achieves high precision and recall across all relations. On REBEL Clean, a manually annotated subset of REBEL, SynthIE's macro-F1 score even exceeds that of the original REBEL data's gold annotations. On Wiki-cIE Text, our highest-quality test set, SynthIE outperforms the previous state of the art (with equal model size) by a substantial margin of 57 and 79 absolute points in micro-F1 and macro-F1, respectively.

Overall, we demonstrate that by exploiting asymmetry between structured outputs $y$ and textual inputs $x$, LLMs can generate high-quality synthetic data to train smaller, specialized models. This way, a task that was previously not solvable by LLMs is now feasible for a small 220M-parameter model. Our code, models, and datasets will be released for future researchers to reuse and extend.

## 2 Background and Related Work

### 2.1 Synthetic Data Generation

Several approaches to data augmentation rely on pretrained language models (PLMs). Early efforts include works that require the pre-existence of a dataset, which is then used to finetune the pretrained generator network (Anaby-Tavor et al., 2020; Papanikolaou and Pierleoni, 2020; Yang et al., 2020; Mohapatra et al., 2020; Kumar et al., 2020).

Recent work focuses on methods not requiring supervision beyond a few demonstrations. Wang et al. (2021) generate synthetic labels by providing unlabeled samples as examples to the LLM. Ye et al. (2022) and Gao et al. (2022) use PLMs to generate data with carefully designed prompts. They evaluate on text classification, question answering, and natural language inference. There are similar procedures for GLUE (Wang et al., 2018) tasks (Meng et al., 2022a), intent classification (Sahu et al., 2022), and question answering (Li et al., 2022). Alternatively, Meng et al. (2022c) tune a PLM on a few demonstrations and use it as a synthetic data generator; and Smith et al. (2022) incorporate prompted PLM for weak-supervision; while Shao et al. (2023) generate synthetic demonstrations to improve the propmting of LLMs; and Honovich et al. (2022) generate synthetic instructions for instruction tuning of LLMs. Contrary to these works, we do not generate labels but

prompt an LLM to perform the reverse task when the reverse task is *easy* for the LLM, which results in high-quality data points. To the best of our knowledge, only Meng et al. (2022b) and Gao et al. (2021) also perform the reverse task by prompting an LLM to generate comments *x* given a sentiment *y*. However, their direct task is simple and can already be solved easily by an LLM, and their *y* is not structured.

## 2.2 Closed Information Extraction

In this work, we exemplify our synthetic data generation approach with closed information extraction (cIE), which is the task of extracting the exhaustive set of facts from natural language, expressible under the relation and entity constraints defined by a knowledge base (KB). More formally, assuming a KB concerning a collection of entities $\mathcal{E}$ and collection of relations $\mathcal{R}$, the goal is to extract the exhaustive set of fact triplets $y_{\text{set}} = \{(s, r, o) \,|\, (s, r, o) \in \mathcal{E} \times \mathcal{R} \times \mathcal{E}\}$ expressed in a given textual input *x*.

Contrary to previous pipeline approaches (Galárraga et al., 2014; Angeli et al., 2015; Chaganty et al., 2017), Josifoski et al. (2022) proposed GenIE, a model approaching cIE in an autoregressive end-to-end formulation. Their results suggest that GenIE is more sample-efficient and achieves state-of-the-art performance, with a substantial improvement over previous work. However, despite being 3x higher than the strongest baseline, GenIE's macro-F1 score is less than 35%. They identify the heavily skewed data distribution, with most relations appearing only a few times in the training data, as the main reason for the low macro score.

Whereas cIE requires the constituent elements of the output triplets to be entities and relations from the KB, the output triplets in open information extraction (oIE) are free-form text. Therefore, cIE is a fundamentally harder task than oIE, closely related to a full spectrum of tasks that are central to the field of information extraction (e.g., entity linking, relation classification, slot filling, etc.). We focus on the task of cIE as a use case for studying the benefits SDG could have for information extraction tasks more generally.

## 2.3 Data as a Core Limitation

As Josifoski et al. (2022) noticed, cIE datasets naturally present large imbalances in relation frequencies. Training on such data results in models that perform well only for the few frequent relations and poorly on the rest. Addressing this issue requires

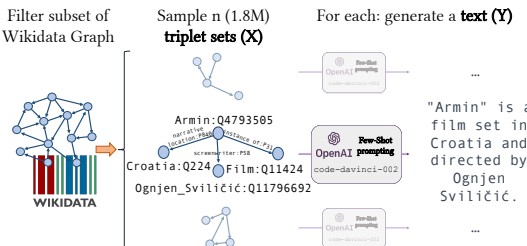

Figure 2: **Synthetic data generation flow.** We start by filtering the Wikidata knowledge graph to a subset of relations and entities comparable to REBEL. Then, we sample coherent triplet sets, encouraging uniform coverage of relations. Finally, we prompt OpenAI LLMs to generate text for each triplet set.

access to more annotated data covering the large number of rare relations which cannot be easily obtained using human annotations or distant supervision heuristics (Huguet Cabot and Navigli, 2021). Instead, our procedure allows us to synthetically generate a large amount of data with good coverage of *all relations*. This data scarcity problem is not unique to cIE, and our procedure can benefit other structured NLP tasks such as entity linking, oIE, or abstract meaning representation parsing (Banarescu et al., 2013), as well as other tasks such as writing assistants (Schick et al., 2022).

## 3 Exploiting Asymmetry for Synthetic Data Generation

In this section, we demonstrate how to exploit asymmetry for synthetic data generation with cIE as an example task. Our pipeline, depicted in Fig. 2, comprises three primary components: (i) construction of a knowledge graph (KG) containing the entities and relations of interest; (ii) sampling of coherent triplet sets from the KG with comprehensive coverage of the entities and relations, and (iii) generation of high-quality text, expressing the triplets without any additional triplets. Next, we describe these three components in turn.

### 3.1 Knowledge Graph Construction

We start from the Wikidata KG (Vrandečić, 2012). To remain comparable to previous work (Josifoski et al., 2022), we filter the KG to the subset of 2.7M entities $\mathcal{E}$ and 888 relations $\mathcal{R}$ appearing at least once in REBEL's training set. Each entity in the KG is associated with a unique English Wikipedia page title, and each relation is linked to a unique Wikidata label, which we use as their textual identifiers (see Appendix B.1 for details).

## 3.2 Sampling Triplet Sets

In the Wikidata KG, nodes represent entities, and edges represent relations between two entities. Therefore, a triplet can be seen as an edge together with its endpoint nodes. For the synthetic data to be valuable, we design the triplet set sampling with two objectives in mind. First, it is crucial that the triplet sets be coherent: the triplets in each set must be able to conceivably co-occur in human-written text. Second, the dataset should have (approximately) uniform coverage of entities and relations.

**Encouraging coherence.** We observed that uniform edge sampling from the KG does not produce coherent triplet sets. Instead, coherent sets of triplets tend to focus on a small number of entities, with one or two entities being repeated in most of the triplets. These entities serve as the *"protagonists",* or anchors, of the sentence. To capture this property, we propose a sampling procedure based on a random walk with backtracking applied to the KG. Concretely, given a starting point—a node (i.e., an entity $e_{\text{sub}}^{(0)}$) or an edge (i.e., a triplet $t_0 = (e_{\text{sub}}^{(0)}, r^{(0)}, e_{\text{obj}}^{(0)})$) from the KG—we maintain a set of already sampled triplets $T$, and until the desired number of triplets is reached, iteratively sample: (i) a subject $e_{\text{sub}}^{(|T|)} \in \mathcal{E}$ starting a new triplet; or (ii) an object $e_{\text{obj}}^{(|T|)} \in N(e_{\text{sub}}^{(|T|)})$, where $N(e)$ corresponds to the set of entities adjacent to $e$, forming a triplet $t_{|T|} = (e_{\text{sub}}^{(|T|)}, r^{(|T|)}, e_{\text{obj}}^{(|T|)})$ to be added to $T$. The entity sampling is biased towards entities already appearing as a subject or an object of a triplet in $T$ by a parameter controlling the strength of the bias. The desired number of triplets per set is sampled from a Poisson distribution. Appendix B.2 details the choice of parameters.

**Encouraging coverage.** When sampling triplet sets from the graph uniformly, some entities and relations are so central that they appear in most local neighborhoods. These end up being over-represented, heavily skewing the distribution.

To alleviate this issue, we implement an aggressive reweighting of the entity and relation distribution. After every $K$ sampled sets, we craft new relation and entity distributions, where the probability of sampling a specific entity or relation is inversely proportional to its frequency in the set of already sampled triplet sets $S$. As a consequence, after each reweighting, the rarest entities and relations are given the highest probability of being sampled. We denote the two distributions as $\mathbb{D}_{\mathcal{E}}^{S}$ and $\mathbb{D}_{\mathcal{R}}^{S}$. Appendix B.2 details the choice of $K$.

**Ensuring coverage.** Rare entities and relations do not often appear in other entities' local neighborhoods. Even if they are associated with a high probability of being sampled when encountered, they are encountered rarely and, therefore, sampled rarely. One way to ensure that rare entities and relations are selected is to explicitly choose them as a starting point of a random walk (i.e., a triplet sets). We describe two strategies for achieving this:

(i) **Entity-centric** strategy: sample the starting entities of each triplet set according to the reweighted entity sampling distribution $\mathbb{D}_{\mathcal{E}}^{S}$.

(ii) **Relation-centric** strategy: sample a relation $r$ according to the reweighted relation sampling distribution $\mathbb{D}_{\mathcal{R}}^{S}$. Then, among the triplets corresponding to relation $r$, sample one according to the probability assigned to the subject entities by the reweighted entity sampling distribution $\mathbb{D}_{\mathcal{E}}^{S}$, renormalized to the ones available.

In both cases, reweighting favors the rarest entities and relations as the next starting points.

For comprehensive coverage of both entities and relations, we employ a **mixed** strategy, which switches between the entity-based and relation-based strategy on every $K$ samples—at the same time when the empirical relation and entity distributions are recomputed.

## 3.3 Triplet-Set-to-Text Generation

In this section, we denote by *query* the triplet set for which we want to generate text. Also, we refer to the in-context examples as *demonstrations*. The demonstrations consist of triplet sets and sentences selected from REBEL's training set. When choosing the triplet-set-to-text generation setup, we made the following considerations.

**LLM choice.** We consider two models from OpenAI's GPT 3.5 series: `code-davinci-002` and `text-davinci-003` (details in Appendix B.3).

**Prompting strategy.** We evaluated both models in a zero-shot and a few-shot setting. In the zero-shot setting, we experimented with different instructions. In the few-shot setting, we varied the instruction, the number of demonstrations, and the formatting of the demonstrations.

**Generation parameters.** We experimented with different values for temperature and top-$p$.

| | min | 1st quartile | median | 3rd quartile | max |
|---|---|---|---|---|---|
| REBEL | 1 | 4 | 34 | 432 | 716,679 |
| Wiki-cIE Code | 65 | 934 | 1380 | 3629 | 479,250 |
| Wiki-cIE Text | 4 | 42 | 62 | 136 | 14,323 |

Table 1: **Relation occurrence count statistics.**

| | | *Micro* | | *Macro* |
|---|---|---|---|---|
| | Precision | Recall | F1 | Recall |
| REBEL | 29.35 ±7.77 | 56.05 ±10.40 | 39.87 ±7.62 | 24.20 ±6.20 |
| Wiki-cIE Code | 57.40 ±10.28 | 70.38 ±7.83 | 65.08 ±7.35 | 50.70 ±9.10 |
| Wiki-cIE Text | 84.78 ±5.80 | 78.45 ±8.20 | 82.97 ±5.53 | 72.14 ±8.73 |

Table 2: **SDG quality (human evaluation) results.**

We ran the inference on a selected set of 12 triplet sets from REBEL's validation set and manually evaluated the quality of the generated outputs in terms of precision, recall, and fluency. The best-performing prompt setup and optimal set of generation parameters for both models are given in Fig. 5 and Table 4, respectively. In Table 5, we showcase example generations for a few data points. In this setup, we generated two datasets:

**Wiki-cIE Code** consists of around 1.8M training, 10K validation, and 50K test samples generated with `code-davinci-002`.

**Wiki-cIE Text** consists of 10K validation and 50K test samples generated with `text-davinci-003` using the same triplet sets as in Wiki-cIE Code. Appendix B.3 contains the inference costs details.

### 3.4 Distributional Properties of Data

One important problem we aimed to address with our synthetic data generation is the imbalance in relation frequencies. Table 1 reports the basic statistics of the relation frequency distribution in REBEL, Wiki-cIE Code and Wiki-cIE Text (see Appendix B.4 for the cumulative distribution plots). While REBEL is very skewed towards few relations appearing most of the time, Wiki-cIE Code has a much more balanced distribution. In particular, the rarest relations in Wiki-cIE Code appears more often than the median relation in REBEL. Josifoski et al. (2022) show that supervised models perform poorly on rare relations. Therefore, we expect Wiki-cIE Code to help supervised models perform well on a much larger subset of relations, which is necessary for exhaustive extraction. In terms of entity coverage, the training split of our Wiki-cIE Code contains 1,805,504 unique entities, compared to the 1,715,922 in REBEL's.

### 3.5 Human Evaluation

**Experimental setup.** We randomly selected 50 data points from REBEL's test set and synthetically generated the text for the corresponding triplet sets following the procedure outlined in Sec. 3.3, with the generation parameters and prompts used to generate the Wiki-cIE Code and Wiki-cIE Text datasets. As a result, two more versions of the (small) dataset, differing only in the textual sequences, were created—one following the Wiki-cIE Code and another following the Wiki-cIE Text text generation procedure. We evaluate the match between the triplet-set-to-text pairs for each dataset by scoring data points in terms of standard precision, recall, and F1 (see Appendix D for definitions). Concretely, we extract the triplets actually expressed in each of the three versions of the text by human annotation and compare them against the corresponding target set (i.e., REBEL's gold triplet set) as ground truth. Appendix B.5 details this process. In this setting, precision corresponds to the fraction of triplets expressed in the text that are present in the target set, and recall to the proportion of triplets in the target set that were expressed in the text.

**Results.** The results are summarized in Table 2. First, the results indicate that the synthetically generated text has substantially higher precision and recall than REBEL's original text, with Wiki-cIE Text reaching 84.8% precision and 78.5% and 72.1% in micro- and macro-recall, respectively. Second, REBEL texts score low in precision (29.4%), suggesting that over 70% of the information in REBEL text is absent from the target set. On the other hand, a micro-recall score of 56%, implies that REBEL's gold annotations are actually wrong 44% of the time. Finally, our datasets' micro and macro scores are much closer than REBEL's, indicating that our datasets have more consistent quality across relations.

## 4 Synthetic Data in Action

In this section, we evaluate the benefits of training on synthetically generated data.

### 4.1 Modeling and Inference

**Model.** Given textual input $x$, our proposed model, SynthIE, autoregressively generates the linearized sequence representation $y$ of the exhaustive set of facts $y_{\text{set}}$ expressed in $x$. The conditional probability (parametrized by $\theta$) assigned to the target set

$y_{set}$ is computed as: $p_\theta(y \mid x) = \prod_{i=1}^{|y|} p_\theta(y_i \mid y_{<i}, x)$. The training consists of maximizing the target sequence's conditional log-likelihood with teacher forcing (Sutskever et al., 2011, 2014), using the cross-entropy loss, and dropout (Srivastava et al., 2014) and label smoothing for regularization (Szegedy et al., 2016). We use the same task formulation and training as GenIE (Josifoski et al., 2022), but unlike GenIE, which employs the BART architecture (Lewis et al., 2020), SynthIE is based on FLAN-T5 (Chung et al., 2022), a choice mainly motivated by the availability of pre-trained checkpoints with different parameter counts.

**Output linearization.** To represent the set of facts $y_{set}$ as a sequence of symbols $y$ compatible with sequence-to-sequence architectures, we introduce two mappings: (i) *fully expanded* (FE) and (ii) *subject-collapsed* (SC), which was used by Huguet Cabot and Navigli (2021). While FE concatenates the textual representation of each triplet in the set to obtain the sequence, SC groups the triplets based on the subject entity and concatenates their grouped representations. For more details and examples, see Appendix F.

**Inference.** At inference time, it would be prohibitively expensive to score every set of triplets in the output space. Instead, we search for the top-$k$ eligible options by using constrained beam search (Sutskever et al., 2014; Josifoski et al., 2022) paired with a strategy for dynamically generating the valid prefixes. More concretely, we enforce a bi-level constraint where (i) the high-level structural constraint asserts that the prefix follows a specific linearization schema, and (ii) lower-level validity constraints (via a pre-computed entity and relation trie) ensure only valid entity or relation identifiers (depending on the given element) are generated.

## 4.2 Experimental Setup

**Knowledge base constraints.** We maintain our world of concern to all the entities and relations from the KG described in Sec. 3.1 corresponding to labels that can be fully tokenized by the model's tokenizer (i.e., tokenized labels do not contain unknown tokens), which rules out around 5% of the entities in the KG. Our final entity catalog contains around 2.6M, and the relation catalog 888 items.

**Datasets.** We differentiate between two data regimes: (i) synthetic and (ii) non-synthetic (distantly supervised). For the synthetic regime, we leverage the datasets generated by the SDG procedure described in Sec. 3. Concretely, we use the larger Wiki-cIE Code for training and testing and the smaller Wiki-cIE Text for testing purposes only. Following the main evaluation setup from Josifoski et al. (2022), we use their version of REBEL for training and testing in the non-synthetic regime (see Appendix C for details).

**Baselines.** To isolate the effect of training on synthetic data, we keep the same architecture and vary the training (and validation) data. We use the *GenIE* identifier to refer to models trained in the non-synthetic and *SynthIE* to refer to models trained in the synthetic regime. See Appendix E for details on the hyper-parameters and the compute time.

**Evaluation metrics.** We evaluate the performance in terms of micro and macro precision, recall, and F1, and report a point estimate with a 95% confidence interval constructed from 50 bootstrap samples. Appendix D formally describes the metrics.

## 4.3 Results

### 4.3.1 Human Evaluation on REBEL

The human evaluation in Sec. 3 uncovers fundamental flaws in REBEL's annotations. Approximately 70% of the information from the text is not included in the "gold" set of triplets, and 45% of the triplets are not expressed in the input text (cf. Table 2). As a result, evaluation on REBEL would provide a highly inaccurate picture of the models' performance. Specifically, triplets missing from the "gold" set lead to an underestimation of true precision, while incorrect triplets in the "gold" set can result in an overestimation of precision and an underestimation of recall. These findings raise serious doubts about the validity of REBEL as an evaluation dataset for cIE. Both of these problems (missing and extra triplets) are addressed by our proposed evaluation datasets (cf. Table 2).

To understand the implications of these limitations, we manually annotate 360 randomly selected samples from REBEL. This process results in a new dataset that we refer to as REBEL Clean. We provide detailed information about the annotation procedure in Appendix G.1.

We first evaluate REBEL's gold triplet sets against the hand-annotated triplet sets, by treating the original ones as if they were the output of a model (referred to as REBEL Gold in Table 3). The original annotations achieve an F1 score of 73.8

|  | **Distant Supervision** | | | **Synthetically Generated** | | | | | |
|  | REBEL Clean | | | Wiki-cIE Text | | | Wiki-cIE Code | | |
|  | Precision | Recall | F1 | Precision | Recall | F1 | Precision | Recall | F1 |
|---|---|---|---|---|---|---|---|---|---|
| ***Micro*** | | | | | | | | | |
| REBEL Gold | 92.71 ±1.73 | 60.68 ±2.85 | 73.76 ±2.20 | – | – | – | – | – | – |
| GenIE T5-base | 76.06 ±3.42 | 51.81 ±3.44 | 62.17 ±3.01 | 49.10 ±0.33 | 26.69 ±0.17 | 34.58 ±0.20 | 41.56 ±0.49 | 23.94 ±0.24 | 30.38 ±0.30 |
| SynthIE T5-base | 53.02 ±5.00 | 43.20 ±3.06 | 48.05 ±3.41 | 92.08 ±0.17 | 90.75 ±0.21 | 91.41 ±0.18 | 79.99 ±0.29 | 70.47 ±0.30 | 74.93 ±0.27 |
| SynthIE T5-base-SC | 59.97 ±4.34 | 30.54 ±2.14 | 40.76 ±2.57 | 92.79 ±0.12 | 90.50 ±0.10 | 91.63 ±0.10 | 81.58 ±0.15 | 69.48 ±0.29 | 75.05 ±0.19 |
| SynthIE T5-large | 68.25 ±4.91 | 54.37 ±3.08 | 61.26 ±3.07 | 93.38 ±0.11 | 92.69 ±0.19 | 93.04 ±0.13 | 82.60 ±0.19 | 73.15 ±0.29 | 77.59 ±0.24 |
| ***Macro*** | | | | | | | | | |
| REBEL Gold | 51.21 ±5.03 | 41.02 ±4.69 | 43.76 ±4.62 | – | – | – | – | – | – |
| GenIE T5-base | 39.36 ±4.68 | 31.46 ±4.24 | 33.33 ±4.07 | 29.82 ±0.67 | 11.14 ±0.15 | 13.94 ±0.17 | 25.78 ±0.85 | 9.81 ±0.10 | 12.12 ±0.12 |
| SynthIE T5-base | 35.57 ±4.82 | 34.05 ±4.47 | 33.13 ±4.44 | 94.10 ±0.15 | 92.42 ±0.17 | 93.05 ±0.11 | 83.76 ±0.36 | 74.05 ±0.45 | 77.91 ±0.42 |
| SynthIE T5-base-SC | 20.07 ±3.26 | 12.82 ±2.65 | 14.65 ±2.70 | 94.35 ±0.19 | 92.39 ±0.20 | 93.15 ±0.15 | 84.32 ±0.32 | 73.57 ±0.41 | 77.88 ±0.34 |
| SynthIE T5-large | 54.11 ±5.26 | 52.01 ±4.64 | 51.04 ±4.76 | 95.27 ±0.22 | 94.95 ±0.13 | 94.99 ±0.12 | 86.43 ±0.25 | 78.78 ±0.27 | 81.95 ±0.22 |

Table 3: **Main results.** Performance of our model SynthIE, the baseline GenIE, and REBEL's target annotations, evaluated on the hand-annotated but biased REBEL Clean, Wiki-cIE Code, and the highest-quality Wiki-cIE Text.

micro and 43.76 macro, which is unsatisfactory for a dataset intended for estimating model performance. Furthermore, the significant gap between micro and macro scores confirms that the quality of original annotations varies greatly across relations.

Next, we evaluate the predictions from our models and the baseline, and observe that, in terms of macro performance, (i) SynthIE T5-large outperforms REBEL Gold; and (ii) SynthIE T5-base is on par with GenIE T5-base. Crucially, the first observation suggests that the predictions of SynthIE T5-large—a model trained on synthetic data generated with our proposed methodology—exhibit higher quality than REBEL's original gold triplet sets. On one hand, this highlights the quality of SynthIE, and on the other hand, it further undermines the credibility of an evaluation on REBEL.

It is worth noting that the REBEL Clean dataset inherits some important problems from REBEL: (i) a high imbalance in terms of the relation occurrence counts, which can be exploited by models like GenIE (trained on REBEL) to achieve strong (micro) performance despite only performing well for few relations, (ii) text often containing information for entities that cannot be resolved.

These findings emphasize the importance of the proposed Wiki-cIE Text as a reliable evaluation dataset for the cIE task. By design, Wiki-cIE Text does not suffer from these issues.

### 4.3.2 Performance Evaluation

On Table 3, we start by noticing that GenIE T5-base achieves an F1 score of 62.2 micro and 33.33 macro on REBEL Clean. However, the model's performance decreases by almost half in terms of micro

and two-thirds in macro F1 on Wiki-cIE Text and Wiki-cIE Code. This is due to several reasons. Crucially, the annotations per relation in GenIE's training dataset, REBEL, are not uniform in terms of representation and quality. As a result, the model performs well on a few relations and badly on the rest. This is exposed by the synthetic datasets, which (i) contain triplets expressing every relation seen at least once in REBEL's training set as opposed to REBEL's test set, which does not cover 300 relations (around a third); and (ii) express all of the relations as uniformly as possible.

Additionally, while GenIE's precision also decreases, F1 performance is particularly affected by the strong decrease in recall. For instance, on Wiki-cIE Text, in comparison to REBEL Clean, the precision drops by 17 (micro) and 10 (macro) absolute points while the recall drops by 25 (micro) and 20 (macro) absolute points. The drop in recall is more pronounced as a consequence of (i) training on an imbalanced dataset (REBEL), with non-exhaustive annotations missing 70% of the information; and (ii) evaluation on a balanced, diverse dataset in which almost 85% of the information from the text is present in the target triplet set (see Sec. 3.5).

On the other hand, SynthIE T5-base, which is trained on data synthetically generated by the proposed methodology, and differs from GenIE T5-base only in terms of the training data, achieves a 91.4 micro, and an even higher 93.1 macro-F1 score on Wiki-cIE Text. Going to the larger SynthIE T5-large model, the performance on both datasets increases from 2 to 4 absolute points.

Finally, the subject-collapsed (SC) linearization decreases the target sequence length (see Fig. 8)

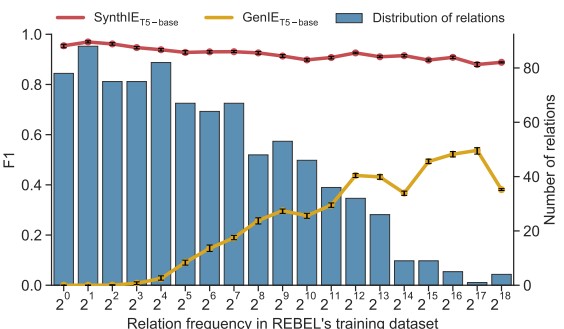

Figure 3: **Impact of the relation frequency.** Relations are bucketed based on their frequency; bucket $2^i$ contains relations occurring between $2^i$ and $2^{i+1}$ times. The histogram shows the number of relations per bucket. The line plots depict the per bucket F1 scores for GenIE and SynthIE evaluated on Wiki-cIE Text, with confidence intervals constructed by bootstrapping.

without any performance costs on Wiki-cIE Text and Wiki-cIE Code. However, the autoregressive nature of the model, paired with the often ill-defined nature of the task in REBEL, renders SC an inappropriate choice for REBEL Clean that comes with a substantial performance cost. This result highlights that the choice of the output format should not be neglected. We further discuss this in Sec. 5.

### 4.3.3 Performance by Relation Frequency

There is a natural imbalance in relation frequencies in text. In existing datasets, most of the triplets correspond to only a few relations (Josifoski et al., 2022). Models trained on such data are good at extracting information concerning a few relations and ignore the rest, which is a major obstacle to exhaustive cIE. For this reason, we bucket relations based on their number of occurrences in REBEL's training set and compute the per-bucket (micro) F1 performance. The results are reported in Fig. 3. For 46% of the relations that have less than $2^5 = 32$ occurrences, GenIE's performance is close to 0. The model's performance slowly starts to rise for relations with at least $2^5 = 32$ occurrences, reaching a maximum of around 58% F1 for the few most frequent relations. Overall, the performance is higher than 50% for buckets that cover only 1.5% of the relations in the dataset. In contrast, training on Wiki-cIE Code, which has a uniform quality and coverage of annotations across relations, makes SynthIE perform well across all buckets. This translates to an F1 performance corresponding to a straight line at the top of the plot, at around 93%.

## 5 Discussion

**Implications for cIE.** The lack of a large, balanced, high-quality dataset has been a significant obstacle for cIE. The synthetic data generated by the methodology proposed in this work satisfies all of the desiderata in terms of size, coverage, and quality (cf. Sec. 3.4 and Sec. 3.5). As an alternative to REBEL which greatly suffers from both false positives and negatives, Wiki-cIE Text enables a substantially more accurate evaluation.

Similarly, Wiki-cIE Code enables the training of models performing well across all relations: (i) in terms of macro performance, SynthIE's predictions are of higher quality than REBEL's *gold annotations*; (ii) on the highest quality test set (Wiki-cIE Text), SynthIE pushes the macro-F1 score of 14%, for GenIE, to 93% (cf. Table 3 and Fig. 3).

While SynthIE's performance on REBEL Clean is better than the original *gold annotations*, it is still lower than the performance on Wiki-cIE Text. Analyzing the errors committed by our models and looking at the REBEL data allowed us to uncover an interesting property of cIE that, to the best of our knowledge, has not been identified by prior work. Assume that a model was trained on exhaustively annotated data (the holy grail which is now within reach) and presented with text containing central information that cannot be linked to the KB, (i) either in theory (e.g., "He plays the guitar") or (ii) under the output constraints assumed by the model. This would place the model in an undesired space of the likelihood distribution where it attempts to express information it cannot express, and is therefore bound to make mistakes. Fortunately, mistakes of this type can be avoided by (i) generating training data covering such scenarios and/or (ii) expanding the expressivity of models by extending the catalogs or developing novel output formats beyond subject, relation, object triplets. The latter goes towards exhaustive IE, whereas the former relaxes this desideratum. Overall, we believe that SDG could bring us closer to practical cIE systems.

**Implications for SDG.** This study highlights the efficacy of leveraging asymmetries in difficulty for SDG by developing a specific pipeline for cIE. However, the idea can be readily applied to different pipelines for cIE, or any other IE or parsing task, such as entity linking, oIE, or abstract meaning representation parsing. With cIE, which necessitates the most information external to the

input (i.e., knowledge of entities, relations, and their IDs), arguably being the hardest task, we are confident that applying our approach to these tasks would translate to similar results.

Finally, the proposed methodology is not limited to tasks with structured output spaces. Rather, any problem with an inverse formulation that can be addressed more effectively by the LLM can benefit from our approach.

## Limitations

**Exhaustive cIE.** We focus on exhaustive cIE, a formulation of the IE task where the model is asked to exhaustively annotate *all the information* expressed in the text. This focus is reflected in the synthetic data generation procedure, where we generate text faithful to the target set. Notably, obtaining a large dataset of comparably faithful text-triplets pairs was not feasible prior to this work. Our results (see Sec. 4.3) suggest that SynthIE, the line of models trained on such high-quality data, can solve the task of exhaustive cIE remarkably well whenever *all the information* in the text *can be* linked to the KB. However, when this is not possible – either due to missing information in the text or limited model expressivity – the models are placed in an undesired space of the likelihood distribution and, consequently, fail to achieve the same performance. As discussed in Sec. 5, we can address this by modifying the data generation procedure or by expanding the expressivity of our models. However, future work would need to decide how important *exhaustiveness* is for *cIE*.

**Model bias.** Synthetic data generation is an integral part of this work. Naturally, relying on an LLM to generate the training data opens the door for the LLM's biases to be introduced into the newly generated synthetic data as well.

**LLM availability.** As described in Sec. 3.3, for the SDG we consider two models from OpenAI's GPT 3.5 series: `code-davinci-002` and `text-davinci-003`. While the API for using `text-davinci-003` is publicly available, `code-davinci-002` is not. However, getting access to it is still possible through OpenAI's Researcher Access Program.

## Acknowledgements

We would like to thank Jiheng Wei, Yifei Li, and Saibo Geng for their help with the human evaluation; Debjit Paul and Veniamin Veselovsky for their helpful feedback on a draft version of the paper; and Akhil Arora for providing us with a mapping of Wikipedia page titles to Wikidata identifiers. West's lab is partly supported by grants from Swiss National Science Foundation (200021_-185043), Swiss Data Science Center (P22_08), H2020 (952215), Microsoft Swiss Joint Research Center, and Google, and by generous gifts from Facebook, Google, and Microsoft.

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

## A LLM cIE failure cases

In Fig. 4 we showcase examples of LLMs not being able to solve the problem of cIE effectively. While `text-davinci-003` can often identify the core information in the text, it is not able to map the names to the Wikidata entities or relations.

## B Synthetic Data Generation

In this section, we give details on the sampling and the synthetic data generation process. Additionally, we provide examples of prompts used to generate the data in Fig. 5, as well as generated sentences with `text-davinci-003` and `code-davinci-002` in Table 5.

### B.1 Details about the Knowledge Graph

We first select entities and relations which appear in the REBEL dataset as described in Sec. 3.1 and relations that do not take literal arguments. We filter out all the entities whose names cannot be associated with a Wikipedia page. This subset comprises 2,715,483 entities and 888 relations. We also exclude entities with a degree of 0, as they do not contribute to any facts in the graph. As a result, our filtered graph includes 2,715,483 nodes and 17,655,864 edges. It is worth noting that our synthetic data generation approach can easily be applied to larger subsets of the Wikidata KG or even the full KG. However, we subsample the KG to remain comparable to previous research.

### B.2 Triplet Sampling

To sample a coherent triplet set, we follow the iterative procedure explained in Sec. 3.2. The parameters involved in this procedure: (i) the *number of triplets* per triplet sets and (ii) the *bias factor*. We sample the number of triplets from a Poisson distribution of mean 3. Assuming a triplet set comprising $N$ different entities, the next (subject or object) entity is sampled from a probability distribution where entities: (i) not appearing in the triplet set are sampled with probability proportional to 1; (ii) entities already appearing in the triplet set are assigned a probability proportional to $(N + 1 - r)^{bf}$, where $bf$ is the bias factor, and $r$ is the rank of the current triplet. To choose the bias factor, we fix a random seed for starting nodes and apply the sampling procedure described above for varying choices of bias factor ([1, 3, 7, 10]). We manually inspect the result triplet set and judge their coherence. We find that 7 and 10 yield good coherence

| parameter | code-davinci-002 | text-davinci-003 |
|---|---|---|
| max_tokens | 100 | 50 |
| temperature | 0.7 | 0.7 |
| top-p | 1 | 1 |
| frequency_penalty | 0.2 | 0.2 |
| presence_penalty | 0 | 0 |
| stop | "\n" | "\n" |
| n | 1 | 1 |
| best_of | 5 | 1 |

Table 4: **Optimal generation parameters for the LLMs used in the SDG.**

but 7 allows for more diversity, whereas 10 always focuses on a single anchor entity. Therefore, we fix the bias factor to 7.

For sampling the starting point, we opt for mixed strategy described in Sec. 3.2. The parameters for this procedure are the dampening factor $d$ of the entity and relation distributions as well as $K$: how often do we recompute the entity and relation distribution from the empirical observation in the current sample (and switch strategy). To choose these parameters, we sample 120K data points with varying $d$ ([0.01, 0.05, 0.1, 0.5, 1]) and $K$ ([2K, 10K, 20K, 120K]). A dampening factor of 1 means no dampening, and $K = N$ the number of samples means no computation of empirical distributions and only using the relation-based strategy. We then look at the skewness, entropy and median number of appearance of relations among the 120K sampled triplet sets, as well as the number of entities covered. We found that a good compromise is given by $d = 0.01$, $K = 20K$.

### B.3 Triplet Set to Text

**Choice of LLM.** The `code-davinci-002` model has been trained on a mix of language and code with further instruction finetuning. `text-davinci-003` was further finetuned with reinforcement learning from human feedback making it more effective at zero-shot learning with instructions but less capable of in-context learning. We query the models through the OpenAI API.

**Inference costs.** At the time of writing, `code-davinci-002` was free with a limitation of 20 requests and 150k tokens per minute. Therefore, the cost for constructing the Wiki-cIE Code dataset was $ 0. `text-davinci-003` was priced at $ 0.02 per 1K tokens, and the total cost of constructing the Wiki-cIE Text was $ 223.55.

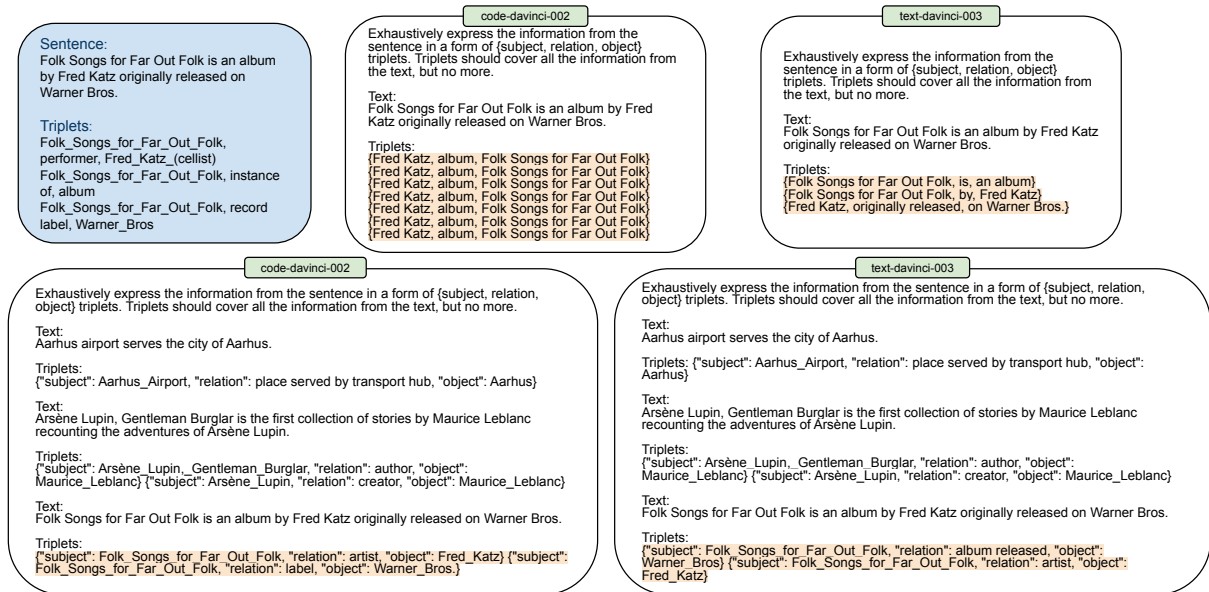

Figure 4: **Examples of failure cases of LLMs attempts to solve cIE task.** In some cases, models are not able to recognize all the facts present in the sentence. Even when this is possible, they are not able to map subjects, relations, and objects to Wikidata concepts.

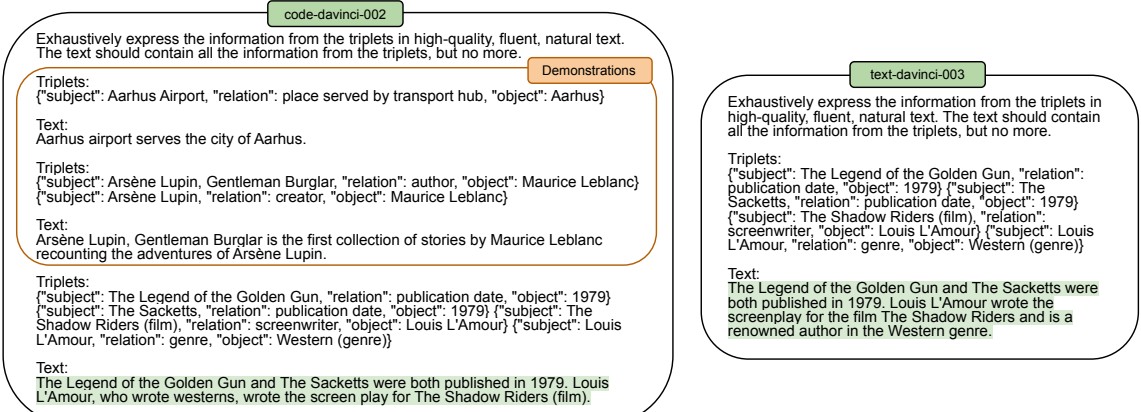

Figure 5: **Best performing prompts.** We present the best-performing prompts for both models, `text-davinci-003` and `code-davinci-002`. `code-davinci-002` makes use of demonstrations. Text highlighted in green corresponds to the output of the model.

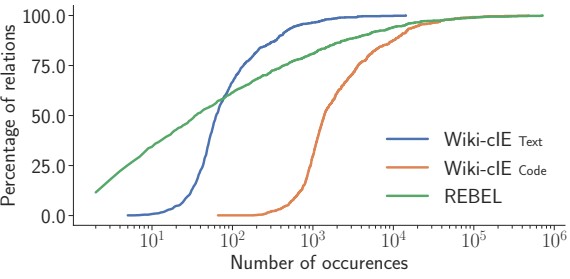

Figure 6: **Cumulative distribution function (CDF) plot of the relation frequencies in each dataset.** The relation frequencies in Wiki-cIE Code and Wiki-cIE Text have a similar CDF graph — follow a similar distribution — shifted on the x-axis due to the difference in the dataset size. In contrast, the relation frequency distribution in REBEL is heavily skewed, with most relations having few occurrences. Despite being larger than Wiki-cIE Code, more than half of the relations in REBEL have fewer occurrences than the least frequent relation in Wiki-cIE Code.

## B.4 Distributional Properties of the Data

In Fig. 6, we showcase the cumulative distribution function of the relation frequencies in REBEL, Wiki-cIE Code and Wiki-cIE Text, providing a more complete view of how REBEL differs from the synthetically generated datasets on this important dimension.

## B.5 SDG Evaluation

### B.5.1 Computing Performance Metrics

To compute the precision and recall (cf. Appendix D), we need the number of: (i) target triplets; (ii) correctly predicted triplets; (iii) triplets expressed in the text (i.e., the total number of predicted triplets).

**Number of target triplets.** This number can be trivially calculated by simply counting the number of triplets in the target set, which comes from REBEL.

**Number of correctly predicted triplets.** We estimate this quantity by hiring Amazon Mechanical Turk (MTurk) workers to annotate the data. They are presented with a sentence and a set of triplets and asked to select those actually expressed in the sentence. A more detailed description of the task is provided in Appendix B.5.2. Appendix B.5.3 details the quality checks and the inter-annotator agreement for the task. In conclusion, this procedure leaves us with the number of correctly pre-

dicted triplets and allows us to compute the micro- and macro-recall.

**Number of triplets expressed in the text.** For this analysis, we aim to exhaustively annotate all the information in the text, even that which is not contained in the triplet set. However, doing this properly requires a non-trivial understanding of the closed information extraction task. Because of that, we opted to estimate this number with three of the authors acting as annotators. Appendix B.5.4 contains the details of the annotation procedure. Given the number of total triplets in the text, we can calculate the micro-precision using the standard definition. This estimate of the precision is, in fact, a (very) tight upper bound on the true precision for each model, as there may exist triplets that were accidentally missed by the annotators. Finally, we note that to enable exhaustive annotation, we relaxed the catalog constraints on the relations in the annotated triplets, which prevented us from computing the macro-precision.

### B.5.2 Human Annotation Task

For each annotation task, we presented the MTurk workers with a single text. Along with the text, a list of potential triplets was provided, and the workers were instructed to indicate which of the triplets were present in the text. In the instructions, we included an example text, a set of triplets paired with explanations why each should or should not be marked, as depicted in Fig. 7.

### B.5.3 Quality of annotations

To ensure high-quality annotations, we took the following measures:

- **Crowdworkers criteria.** In an attempt to get reliable ratings, we recruited workers that have previously completed at least 1000 tasks with 99% acceptance rate. Workers were restricted by location to US, UK and Canada. We targeted a pay rate of $ 8-10 per hour, guided by US minimum wage.

- **Honeypots.** Each MTurk task consisted of 10 rating tasks. Among the 10 rating tasks, there were 1-2 honeypots, each constructed by randomly selecting two different REBEL samples and pairing the sentence from the first sample with the target set from the second. In this case, workers were expected not to mark any of the given triplets, as none of them were

**Instructions**

You will be given a sentence and a list of facts that the sentence possibly expresses. Your task is to mark which of the listed facts are explicitly expressed in the sentence. If none of the facts are expressed in the sentence, you shouldn't select any of the options.

Facts are given in the format (subject; relation; object). For example, the sentence "Joe Biden was born in Pennsylvania" expresses the fact (Joe Biden; place of birth; Pennsylvania).

Figure 7: **Mturk setting.** Workers are presented with the sentence and a list of triplets. Their task is to decide which triplets are present in the presented sentence. They are also presented with detailed instructions and examples.

expressed in the sentence. This allowed us to filter out unreliable workers.

- **Multiple annotators.** Each task was done by three different workers. The final set of triplets expressed in the sentence was constructed by considering the majority vote for each of the triplet in the set. The Fleiss' kappa was 0.2239.

### B.5.4 Precision Annotation Procedure

The annotation was performed in two steps. In the first step, one of the annotators annotated every sentence with the triplets expressed in the text external to the original triplet set. In the second step, two other annotators independently decided whether each annotated triplet was indeed expressed in the generated text. Conflicts were resolved via discussion. This was done without knowing whether the text-triplets pair came from REBEL or one of our synthetically generated datasets. The estimated prediction set was constructed as the union of the original triplet set and the missing triplets that the annotators provided. 6 data points were excluded from the final analysis due to problems with the original data.

### C Datasets

**REBEL** Following the processing in Josifoski et al. (2022), we adapt REBEL (Huguet Cabot and Navigli, 2021) for close information extraction by linking entities to their corresponding Wikipedia page and filtering out those that do not have an associated Wikipedia page (i.e., entities not associated to a unique name). The original dataset is created from Wikipedia abstracts. It consists of

an alignment between sentences, Wikipedia hyperlinks, and their corresponding Wikidata entities and relations. REBEL proposed an alignment expanding on Elsahar et al. (2018), a pipeline of mention detection, coreference resolution, entity disambiguation, and then mapping triplets to each sentence. Huguet Cabot and Navigli (2021) further filtered false positives using a natural language inference model to check if the relation was truly entailed by the text. We use the processed version of the dataset for training and testing. Sec. 3.4 contains additional statistics for the dataset.

### D Performance Metrics

We measure standard precision, recall, and F1 for all settings. A fact is regarded as correct if the relation and the two corresponding entities are all correct. More precisely, we denote the set of all predicted triplets of a document $d \in \mathcal{D}$ as $P_d$, and the set of gold triplets as $G_d$. Then:

$$\text{micro-precision} = \sum_{d \in \mathcal{D}} |P_d \cap G_d| \Big/ \sum_{d \in \mathcal{D}} |P_d|, \quad (1)$$

and

$$\text{micro-recall} = \sum_{d \in \mathcal{D}} |P_d \cap G_d| \Big/ \sum_{d \in \mathcal{D}} |G_d|. \quad (2)$$

Micro scores are useful for measuring the overall performance of a model, but they are less informative for imbalanced datasets (e.g., when some entities or relations are disproportionately more present in both training and test sets). Indeed, micro scores assign equal weight to every sample, while macro scores assign equal weight to every class. For this reason, we also measure and report

| triplets | rebel text | text-davinci-003 | code-davinci-002 |
|---|---|---|---|
| (Samsung Galaxy Xcover 3; operating system; Android (operating system)), (Samsung Galaxy Xcover 3; manufacturer; Samsung Electronics) | The Samsung Galaxy Xcover 3 is an Android smartphone produced by Samsung Electronics and released in July 2015. | **The Samsung Galaxy Xcover 3 is a smartphone manufactured by Samsung Electronics that runs on the Android operating system.** | **The Samsung Galaxy Xcover 3 is a smartphone manufactured by Samsung Electronics that runs the Android operating system.** |
| (Cleveland Cavaliers; league; National Basketball Association) | During his freshman year he helped lead them to a West Coast Athletic Conference championship and a berth into the 1985 NCAA Tournament, where Pepperdine would lose in the first round. Middlebrooks was named the WCAC Freshman of the Year. The following season, the Waves once again won the WCAC, qualified for the 1986 NCAA Tournament, and lost in the first round. This would be Middlebrooks' last experience with the NCAA Tournament as the Waves did not make it during his junior season and only qualified for the 1988 National Invitation Tournament in his senior year. | **The Cleveland Cavaliers are a professional basketball team that plays in the National Basketball Association (NBA).** | The Cleveland Cavaliers are a professional basketball team based in Cleveland, Ohio. They compete in the National Basketball Association (NBA) as a member of the league's Eastern Conference Central Division. |
| (Pix Brook; located in the administrative territorial entity; Hertfordshire), (Pix Brook; mouth of the watercourse; River Hiz), (Hertfordshire; shares border with; Bedfordshire), (Bedfordshire; shares border with; Hertfordshire), (River Hiz; tributary; Pix Brook) | Pix Brook is a stream that flows through Letchworth Garden City in Hertfordshire, Stotfold in Bedfordshire, and meets the River Hiz north of Arlesey. | Pix Brook is located in the administrative territorial entity of Hertfordshire, and is the mouth of the watercourse River Hiz, which is a tributary of Pix Brook. Hertfordshire shares a border with Bedfordshire, and likewise Bedfordshire shares a border with Hertfordshire. | **Pix Brook is a tributary of the River Hiz and is located in Hertfordshire, which borders with Bedfordshire.** |
| (1999 Nunavut general election; point in time; 1999) | David Iqaqrialu (born July 2, 1954) is a former territorial level politician from Clyde River, Northwest Territories (now Nunavut), Canada. He served as a member of the Nunavut Legislature from 1999 until 2004. | **The 1999 Nunavut general election took place in 1999.** | **The 1999 Nunavut general election was held in 1999.** |
| (Poltava Governorate; country; Russian Empire) | Andriy Bobyr was born in the village of Nychyporivka, in the Poltava Governorate of the Russian Empire. | **The Poltava Governorate was part of the Russian Empire.** | **Poltava Governorate was part of the Russian Empire.** |
| (United States; contains administrative territorial entity; Wisconsin), (Wisconsin; located in the administrative territorial entity; United States) | This genus contains six species that are distributed across the Eastern continental United States as far west as Oklahoma and Wisconsin. | The United States contains the administrative territorial entity of Wisconsin, which is located within the United States. | **Wisconsin is a state in the United States.** |
| (Two Weeks with the Queen; publication date; 1990), (Two Weeks with the Queen; author; Morris Gleitzman) | Two Weeks with the Queen is a 1990 novel by Australian author Morris Gleitzman. | **Two Weeks with the Queen, a novel by Morris Gleitzman, was first published in 1990.** | **Morris Gleitzman's book "Two Weeks with the Queen" was published in 1990.** |
| (Ciudad del Este; country; Paraguay) | Antonio Oddone Sarubbi", is a football stadium in the city of Ciudad del Este, Paraguay. | **Ciudad del Este is located in Paraguay.** | **Ciudad del Este is a city in Paraguay.** |

Table 5: **Text comparison for REBEL triplet sets** This table contains the original REBEL data, as well as text generated using two OpenAI models, `code-davinci-002` and `text-davinci-003`, for the same triplet sets. Synthetic data samples are better in terms of recall, especially precision, and remain fluent. Samples that are overall better in terms of precision, recall, and fluency are bolded.

| Dataset | # Data Points | | | # Triplets | | | # Entities | | | # Relations | | |
|---|---|---|---|---|---|---|---|---|---|---|---|---|
| | train | val | test | train | val | test | train | val | test | train | val | test |
| REBEL | 2,813,210 | 155,926 | 156,449 | 7,187,915 | 397,326 | 398,252 | 2,038,741 | 205,080 | 205,549 | 1071 | 691 | 690 |
| REBEL* | 2,069,780 | 114,448 | 114,953 | 4,642,624 | 256,327 | 257,129 | 1,537,472 | 151,617 | 151,997 | 865 | 595 | 580 |
| Wiki-cIE Code | 1,815,378 | 10,000 | 50,286 | 6,055,911 | 34,262 | 172,991 | 1,806,126 | 27,553 | 105,176 | 888 | 883 | 888 |
| Wiki-cIE Code* | 1,669,708 | 9,222 | 46,210 | 5,482,658 | 31,073 | 156,350 | 1,668,198 | 25,295 | 96,337 | 876 | 869 | 876 |
| Wiki-cIE Text | – | 10,000 | 50,286 | – | 34,262 | 172,991 | – | 27,553 | 105,176 | – | 883 | 888 |
| Wiki-cIE Text* | – | 9,230 | 46,295 | – | 31,117 | 156,805 | – | 25,323 | 96,483 | – | 869 | 876 |

Table 6: **Statistics of the datasets.** *The filtered version of the dataset used in this work. We filter out data points that have: (i) triplets in their respective target set corresponding to entities and relations outside of the pre-defined knowledge base constrained (cf. Sec. 4.1); (ii) input longer than 256 tokens; (iii) linearized output longer than 256 tokens (always according to the longer fully expanded linearization schema in order to keep the same data points across runs).

performance in terms of macro scores. If we denote $P_d^{(r)}$ and $G_d^{(r)}$ as the predicted and gold set only containing the relation $r \in \mathcal{R}$ of a document $d$, then macro-precision is defined as:

$$\frac{1}{|\mathcal{R}|} \sum_{r \in \mathcal{R}} \left( \sum_{d \in \mathcal{D}} |P_d^{(r)} \cap G_d^{(r)}| \Big/ \sum_{d \in \mathcal{D}} |P_d^{(r)}| \right), \quad (3)$$

and macro-recall as:

$$\frac{1}{|\mathcal{R}|} \sum_{r \in \mathcal{R}} \left( \sum_{d \in \mathcal{D}} |P_d^{(r)} \cap G_d^{(r)}| \Big/ \sum_{d \in \mathcal{D}} |G_d^{(r)}| \right). \quad (4)$$

# E Experiment Implementation Details

**Data..** The train, test, and validation splits are inherited for REBEL (Huguet Cabot and Navigli, 2021) and sampled at random for the newly released datasets. We restrict the data points to those with input and target sequences with at most 256 tokens. To facilitate reproducibility, we release the exact splits used in our experiments.

**Training.** The models were trained using the Adam optimizer with a learning rate of 3e-4, 0.1 gradient clipping on the Euclidean norm, and a weight decay of 0.05. We trained the models for 8000 steps, with a batch size of 2568, and a polynomial learning rate scheduler with 1000 warm-up steps and a final learning rate of 3e-05. Following the hyper-parameter optimization for GenIE (Josifoski et al., 2022), we used the default values for most of the parameters and tuned: the number of training and warm-up steps, the batch size, and the weight decay. Importantly, due to the high training costs, the parameters were tuned on the REBEL dataset (the GenIE models), and the best-performing set of parameters was reused for training on the synthetic datasets (the SynthIE models). Having a

better-optimized set of hyper-parameters set gives an advantage to the baseline models, which we expect to be insignificant. The complete parameter configuration and the code to reproduce the experiments is available in the GitHub repository provided in the abstract.

**Inference.** Following Josifoski et al. (2022), we use Constrained Beam Search with 10 beams. We normalize the log probabilities by sequence length and allow for any number of n-gram repetitions. Additionally, we experimented on the validation set with the value of the length penalty parameter. The value of 0.8 was optimal for the models with fully-expanded linearization and 0.6 for the models with subject collapsed linearization. The other parameters are kept to their default values.

**Infrastructure and compute time.** For training all of the models except for the SynthIE T5-large, we used a single machine with 24 Intel(R) Xeon(R) CPU E5-2690 v4 @ 2.60GHz processor cores and 441 GB of RAM, equipped with 4 Tesla V100-PCIE-16GB GPUs. Each training run took 40-45 wall-clock hours (160-180 GPU hours), and each inference run on REBEL took around 16 wall-clock hours (64 GPU hours) and 6 wall-clock hours (24 GPU hours) on the Wiki-cIE Code and Wiki-cIE Text datasets.

The SynthIE T5-large model was trained on a machine with 96 Intel(R) Xeon(R) CPU @ 2.20GHz processor cores and 680 GB of RAM, equipped with 8 Tesla A100-PCIE-40GB GPUs. A training run, in this case, took around 30 wall-clock hours (240 GPU hours), and an inference run took around 11 and 4 wall-clock hours (88 and 24 GPU hours) on REBEL, and the Wiki-cIE Code or Wiki-cIE Text datasets, respectively.

## F Output Linearization

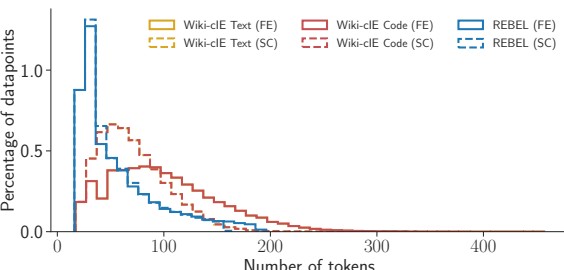

| Triplet Set | (Mount Lanning; instance; Mountain), (Mount Lanning; mountain; Sentinel Range), (Newcomer Glacier; mountain; Sentinel Range) |
|---|---|
| **Fully Expanded** | [s] Mount_Lanning [r] instance of [o] Mountain [e] [s] Mount_Lanning [r] mountain range [o] Sentinel_Range [e] [s] Newcomer_-Glacier [r] mountain range [o] Sentinel_Range [e] |
| **Subject Collapsed** | [s] Mount_Lanning [r] instance of [o] Mountain [e] [r] mountain range [o] Sentinel_Range [e] [s] Newcomer_Glacier [r] mountain range [o] Sentinel_Range [e] |

Table 7: **Example for the different linearization methods.** In the first row, we showcase the original triplet set. The following two rows show the linearization of the original triplet set according to the two methods we consider: fully expanded and subject collapsed.

Figure 8: **Histogram of the number of output tokens according to different linearization strategies.** The subject-collapsed (SC) linearization results in shorter sequences – an effect which is particularly pronounced for Wiki-cIE Code and Wiki-cIE Text where the triplet sets are more exhaustive.

In Sec. 4.1, we introduced two output linearization methods: (i) *fully expanded* (FE) and (ii) *subject collapsed* (SC). This section expands on the two methods. Table 7 presents example outputs.

**Fully expanded linearization (FE).** FE mapping, used by GenIE, starts by linearizing each (subject, relation, object) triplet by using the delimiters [s], [r], [o] to demarcate the start of the subject entity, the relation type, and the object entity, respectively. The end of each triplet is demarcated with [e]. The final representation $y$ is constructed by concatenating the textual representations of all of the triplets in the set $y_{set}$. The advantage of this method lies in its simplicity. However, with most textual sequences introducing several facts per entity (most concerning the same subject), many entities are repeated. To alleviate this, we consider the SC mapping.

**Subject collapsed linearization (SC).** SC mapping, used by Huguet Cabot and Navigli (2021), starts by grouping all of the triplets based on the subject and then linearizing them separately by expressing the group's subject once and then listing the relation and object for each of the triplets in alternating fashion, demarcating the start of each part with the previously introduced delimiters. The end of each triplet group is demarcated by the same delimiter [e], and the final representation is constructed by concatenating the textual representations of all of the triplet groups. The SC linearization results in shorter target sequences at the cost of more complex subpart dependencies in the output. Figure Fig. 8 captures this effect for

the datasets considered in this work. This effect is particularly pronounced for Wiki-cIE Code and Wiki-cIE Text where the triplet sets are more exhaustive, leading to repetitions. For REBEL, which is missing many triplets from the target set the difference in output length between the two linearization methods is not substantial.

While the sequence representation has an intrinsic notion of order, the output set of triplets does not. To mitigate the effects of this discrepancy, we enforce a consistent ordering of the target triplets during training by considering first the triplets corresponding to subjects appearing earlier in the sentence. Ties are resolved by the appearance position of the object entity. Whenever the triplets' entities are not linked to entity mentions in the textual input, we use a heuristic that links each entity to the largest sequence of words from the textual input appearing in the entity name (or to the beginning of the sentence in case of no overlap).

## G SynthIE in Action

### G.1 Human Evaluation of REBEL

#### G.1.1 Constructing REBEL Clean

**Choice of data points.** We started by randomly selecting 1000 data points from REBEL's test set, ordering them according to their corresponding numeric identifier (ID), and printing their ID and input text (crucially, *without* looking at the target triplets). By manually inspecting them in order, we selected data points until we reached 360.[1] We used two simple criteria for choosing the data points.

Criterion 1: The text should have substantial "extractable" information.

One of the samples that were discarded due to this criterion is "In addition to saxophone, he plays

---

[1] 350, and 10 as a backup in case of annotation issues, but we did not encounter any issues and kept all of them.

clarinet, bass clarinet, French horn, flute, and cornet.".

Rationale: Without extractable information, we would be spending our "sample budget" on examples that cannot be resolved even in theory and are, therefore, misleading and not informative.

Criterion 2: The central information in the text should not be of a literal type.

One of the samples that were discarded due to this criterion is: "Incumbent Republican Alfred E. Driscoll defeated Democratic nominee Elmer H. Wene with 51.54% of the vote."

Rationale: GenIE (Josifoski et al., 2022) does not support literals. While extending GenIE (and therefore SynthIE) to literals is possible, that is not the focus of our work. To isolate the effect of synthetic data, we choose to keep the same setting as Josifoski et al. (2022) and limit our SDG to relations that do not rely on literal arguments. Therefore, following the same reasoning as in Criterion 1, data points for which the central information involves literals would be misleading and not informative.

**Construction of target-triplet candidate sets.** It is relatively straightforward and accurate to establish which of the triplets in a given set are expressed in a specific text by considering whether each of them is expressed separately (see Appendix B.5). However, providing workers with our entity catalog containing around 2.6M and the relation catalog of 888 items and asking them to annotate the sentence exhaustively is a difficult task that is bound to result in inaccurate annotations. To circumvent this issue, instead of asking annotators to annotate a sentence exhaustively, we cast the task of "almost" exhaustive annotation in the same format as the task of estimating precision by providing the annotators with a larger "target set" of candidate triplets (with high recall) and ask them to establish which of triplets are indeed expressed in the text. This process will be as exhaustive as the recall of the candidate set, i.e., we will correctly detect the triplets within the candidate set that are expressed in the text and overlook any triplets outside of the candidate set.

In the construction of REBEL Clean, for each datapoint, we construct the candidate target set as the union of the triplets in the corresponding target set in REBEL and the set of predicted triplets by SynthIE T5-large. The rationale here is as follows.

We start from the assumption that any candidate target set should be a super-set of REBEL's original target set, which we know is not exhaustive. Therefore, to increase the coverage, we enlarge the target set in REBEL with the predictions of our best model SynthIE T5-large, which is trained to provide exhaustive annotations. The human annotators will then filter out any incorrectly annotated triplets in the candidate target set.

This design of the human-annotation procedure ensures that (i) our estimate of the true precision will be correct (up to human error) and (ii) the measured recall will be an upper bound with a constant multiplicative gap to the true recall for all the models of interest (equal to the portion of triplets that are potentially missing from the curated target set). We discuss these consequences in Sec. G.1.2 and Sec. G.1.3

**Human annotation task.** This annotation task followed the same format as the task described in Appendix B.5.2. However, to ensure the highest possible quality, for this task, instead of relying on MTurk workers, we hired two Ph.D. and two MSc students, who were not familiar with our work (to avoid potential bias). For their time, they were compensated 25 CHF per hour.

We conducted the annotation in two stages. In the first stage, one of the Ph.D. students annotated all of the data points, while each MSc student annotated half of the data points such that every data point was annotated twice. In the second stage, the second Ph.D. student annotated each data point where the two annotations (from the first Ph.D. student and from one of the MSc students) did not match, resolving the conflicts.

### G.1.2 Precision on REBEL Clean

To compute the precision (see Appendix D), we need the number of: (i) predicted triplets; and (ii) correctly predicted triplets.

The total number of predicted triplets across all documents $d \in \mathcal{D}$, by the model $m$, is given as:

$$P_m = \sum_{d \in \mathcal{D}} |P_{d,m}| \tag{5}$$

where $P_{m,d}$ is the set of triplets predicted by model $m$ for document $d$. This number can be trivially calculated for any model and dataset.

Let $G_d$ be the gold set of triplets in the REBEL Clean dataset and $G_d^* = G_d \cup U$ the *true* set of target triplets where $U$ corresponds to the

set of triplets in the text that are missing from $G_d$. With this notation, the total number of correctly predicted triplets across all documents $d \in \mathcal{D}$, by the model $m$, is defined as:

$$C_m^* = \sum_{d \in \mathcal{D}} |P_{m,d} \cap G_d^*| \qquad (6)$$

Since $G_d^*$ is unknown in practice, to estimate precision, we rely on

$$C_m = \sum_{d \in \mathcal{D}} |P_{d,m} \cap G_d| . \qquad (7)$$

However, in the construction of the gold set of triplets in REBEL Clean, for document $d$, the human annotators considered the triplets $A_d = P_{d,REBEL_{Gold}} \cup P_{d,SynthIE\ T5-large}$ where $P_{REBEL_{Gold}}$ corresponds to the gold (target) set of triplets in REBEL (as the predictions of a model $REBEL_{Gold}$) and $P_{SynthIE\ T5-large}$ to the predictions from SynthIE T5-large. As a consequence, for $REBEL_{Gold}$ and SynthIE T5-large the estimated and the true total number of correctly predicted triplets, and therefore the estimated and the true micro- and macro-precision will be equivalent (up to human error). More generally, for any model $m$, if:

$$\sum_{d \in \mathcal{D}} |P_{m,d} \cap G_d^*| \approx \sum_{d \in \mathcal{D}} |P_{m,d} \cap A_d \cap G_d^*| \qquad (8)$$

the true precision will be estimated well by the standard precision on the REBEL Clean. Considering that GenIE was trained on REBEL, and SynthIE T5-base is a smaller version of SynthIE T5-large, this criterion should be satisfied, and therefore, we expect to have good estimates of micro- and macro-precision for these models as well.

### G.1.3 Recall on REBEL Clean

The previous section argues that, for all of the models of interest, $C_m \approx C_m^*$. As a consequence, using the same notation as the previous section, for the computation of the micro-recall (see Appendix D) of model $m$, we have:

$$
\begin{aligned}
\text{micro-R}_m^* &= \frac{C_m^*}{\sum_{d \in \mathcal{D}} |G_d^*|} \\
&\approx \frac{C_m}{\sum_{d \in \mathcal{D}} |G_d^*|} \\
&= \frac{C_m}{\sum_{d \in \mathcal{D}} |G_d|} \times \frac{\sum_{d \in \mathcal{D}} |G_d|}{\sum_{d \in \mathcal{D}} |G_d^*|} \\
&= \text{micro-R}_m \times \text{micro-R}_{REBEL\ Clean}^*
\end{aligned}
$$

A similar argument can be made for the macro-recall (see Appendix D for the formal definition). Let $C_m^{(r)}$, $C_m^{*(r)}$, $G_d^{(r)}$, $G^{*(r)}$ denote their respective quantities as before but computed on the subset of triplets corresponding to relation $r \in \mathcal{R}$. Then, for model $m$, we have:

$$
\begin{aligned}
\text{macro-R}_m^* &= \frac{1}{|\mathcal{R}|} \sum_{r \in \mathcal{R}} \frac{C_m^{*(r)}}{\sum_{d \in \mathcal{D}} |G_d^{*(r)}|} \\
&\approx \frac{1}{|\mathcal{R}|} \sum_{r \in \mathcal{R}} \frac{C_m^{(r)}}{\sum_{d \in \mathcal{D}} |G_d^{*(r)}|} \\
&= \frac{1}{|\mathcal{R}|} \sum_{r \in \mathcal{R}} \frac{C_m^{(r)}}{\sum_{d \in \mathcal{D}} |G_d|} \times \frac{\sum_{d \in \mathcal{D}} |G_d^{(r)}|}{\sum_{d \in \mathcal{D}} |G_d^{*(r)}|} \\
&\approx \frac{1}{|\mathcal{R}|} \sum_{r \in \mathcal{R}} \frac{C_m^{(r)}}{\sum_{d \in \mathcal{D}} |G_d|} \times \frac{\sum_{d \in \mathcal{D}} |G_d|}{\sum_{d \in \mathcal{D}} |G_d^*|} \\
&= \text{macro-R}_m \times \text{micro-R}_{REBEL\ Clean}^* .
\end{aligned}
$$

However, the equivalence, in this case, relies on an additional assumption used in the penultimate equality: the recall of the ground truth annotations in REBEL Clean should be approximately the same across all relations.