# OpenReview forum: "Exploiting Asymmetry for Synthetic Training Data Generation: SynthIE and the Case of Information Extraction"
_EMNLP/2023/Conference — EMNLP 2023 Main_

### Official Review · Reviewer_kAnw · 2023-08-04

**Soundness:** 4

**Excitement:**

4: Strong: This paper deepens the understanding of some phenomenon or lowers the barriers to an existing research direction.

**Paper Topic And Main Contributions:**

The author proposes utilizing a large model to generate data for training, thereby circumventing the issue of structured output in information extraction. The paper takes relation extraction as an example and conducts a comparative analysis of the quality of synthetic data and human-generated data. In conclusion, it is demonstrated that large models are capable of generating high-quality data suitable for training purposes.

**Reasons To Accept:**

The paper uses relation extraction as a case study and synthesizes data based on the REBEL dataset. In the experimental section, the author conducts a thorough comparative analysis between the generated data and the original dataset, thus validating the effectiveness of the method. The paper provides comprehensive and detailed data, ensuring a valid analysis.

**Reasons To Reject:**

The generated data completely relies on appropriate templates and the generation capabilities of LLMs, which may have problems (like noise) in some week models.

**Reproducibility:**

4: Could mostly reproduce the results, but there may be some variation because of sample variance or minor variations in their interpretation of the protocol or method.

**Reviewer Confidence:**

4: Quite sure. I tried to check the important points carefully. It's unlikely, though conceivable, that I missed something that should affect my ratings.

---

> ### Author Rebuttal · Authors · 2023-08-29
>
> We would like to thank the reviewer for the positive and encouraging review.
>
> Indeed, we acknowledged in the limitations section that as a method for generating synthetic data that leverages an LLM, its effectiveness will be affected by the capabilities as well as biases of the specific LLM used in the process. However, considering the capabilities of today’s LLMs and the effectiveness of the proposed method on the task of closed information extraction, we believe that a carefully designed data-generation procedure, which could potentially be augmented by external knowledge or tools, will readily provide value for many tasks.

---

### Official Review · Reviewer_CoFT · 2023-08-05

**Soundness:** 4

**Excitement:**

3: Ambivalent: It has merits (e.g., it reports state-of-the-art results, the idea is nice), but there are key weaknesses (e.g., it describes incremental work), and it can significantly benefit from another round of revision. However, I won't object to accepting it if my co-reviewers champion it.

**Paper Topic And Main Contributions:**

The paper demonstrates the effectiveness of this synthetic data generation by LLMs on closed information extraction, where collecting ground-truth data is challenging, and no satisfactory dataset exists to date. Authors synthetically generate a dataset of 1.8M data points,
establish its superior quality compared to existing datasets in a human evaluation.

**Reasons To Accept:**

1. The authors propose an effective method to generate large-scale dataset for closed IE tasks.
2. The authors propose many methods for different tasks to generate the dataset.
3. The authors conduct so many experiments and analyze the methods, the paper is solid.

**Reasons To Reject:**

I think generate the dataset only depending on LLMs just like a knowledge distillation. Although the method is effective for smaller model, I think it is useless for LLMs. I think this kind of work does not make me exciting.

**Reproducibility:**

4: Could mostly reproduce the results, but there may be some variation because of sample variance or minor variations in their interpretation of the protocol or method.

**Reviewer Confidence:**

3: Pretty sure, but there's a chance I missed something. Although I have a good feel for this area in general, I did not carefully check the paper's details, e.g., the math, experimental design, or novelty.

---

> ### Author Rebuttal · Authors · 2023-08-29
>
> We are grateful the reviewer highlights the effectiveness of the proposed approach as well as the systematicity of the analysis and the overall soundness of the paper.
>
> In our response, we would like to provide clarifications (reacting to potential misunderstandings) that we believe may further emphasize the benefits of our approach.
>
> **Knowledge distillation and using an LLM *only***\
> In the context of closed information extraction (cIE), as illustrated in Figure 4, the LLM cannot solve the task directly (X → Y) because it has no knowledge of the entity and relation catalogs. However, thanks to the proposed synthetic data generation strategy that exploits an asymmetry in difficulty when going in the reverse direction Y → X, and leverages external information in the form of a knowledge graph in the triplet set sampling, the LLM can provide substantial value, although the direct task is out of reach. The resulting data generation process uses the LLM as a tool and goes way beyond vanilla knowledge distillation.
>
> **The value for LLMs**\
> The paper proposes a general method, which is by no means limited to cIE and can provide value to any IE or parsing task, as well as any task with an inverse formulation that can be addressed more effectively by LLMs. For more details on this, please see our response to R1.
>
> Although we focus on fine-tuning a “small” LM (FLAN-T5) in this paper, our method may be used to further improve large LMs (LLMs) – especially since, as emphasized above and as argued in the paper, there are still many tasks (including cIE, our focus task) that LLMs cannot solve out of the box, but that they can learn to solve based on data as generated by our method. Additionally, our proposed approach could be used for augmenting data used in LLM fine-tuning. Augmentations that could provide value include pairing answers with (i) instructions; (ii) questions and tool calls; or (iii) reasoning traces in domains where the model struggles. These could be useful to improve the model’s performance more generally or to introduce the model to novel tools or/and reasoning patterns.
>
> Outside of LLMs, the proposed method can result in models (LMs, classifiers, etc.) that outperform the original (non-finetuned) LLM at a fraction of the compute cost (e.g., cIE), thereby enabling new use cases or simply cost-efficient solutions. Closed information extraction and tasks suffering from imbalances or a general lack of diversity, if addressed by a general-purpose LLM, would greatly benefit from the proposed methodology (e.g., sarcasm categorization, fine-grained sentiment analysis, etc.).
>
> In conclusion, the proposed method creates many opportunities in the context of LLMs and smaller models alike.

---

### Official Review · Reviewer_qw9q · 2023-08-05

**Typos Grammar Style And Presentation Improvements:** N/A
**Soundness:** 4

**Excitement:**

3: Ambivalent: It has merits (e.g., it reports state-of-the-art results, the idea is nice), but there are key weaknesses (e.g., it describes incremental work), and it can significantly benefit from another round of revision. However, I won't object to accepting it if my co-reviewers champion it.

**Missing References:**

N/A

**Paper Topic And Main Contributions:**

This paper proposed an approach to construct synthetic training data for information extraction using OpenAI's LLM APIs.

Main contributions include:
1. A synthetic data generation approach for information extraction, i.e., first properly sample a triplet from a knowledge graph and then generate a desired text expressing the triplet.
2. Human evaluation of the synthetic data and its counterpart, a human-curated dataset called REBEL. The evaluation demonstrates the high quality of the synthetic data and the important flaws of REBEL.
3. An information extraction model trained on the constructed synthetic data that achieves strong performance.

**Questions For The Authors:**

In Figure 2, it is illustrated that multiple triplets (more than 2 entities and 1 relation) are sampled to generate texts but actually only one triplet is sampled to generate a text each time, right?

**Reasons To Accept:**

1. The paper is well-written and well-organized.
2. Through human evaluation and model performance, the paper shows that model-generated training data can be of higher quality than human-curated data in the field of closed information extraction (cIE). This can be regarded as prior evidence of the population of model-generated datasets.

**Reasons To Reject:**

1. The task, data, and baselines are limited so the audience may be narrow.
2. According to Table 3, the model trained on the synthetic data under-performed baselines on the existing dataset, REBEL. I know there exist some flaws in REBEL's annotation, which are uncovered by the human evaluation, but this does not fully convince me the significant performance loss is acceptable. It would be better to take a closer look at the samples in REBEL that SynthIE failed but GenIE succeed and show that a majority of such samples are contributed by the ill-annotation of REBEL or some other acceptable reasons.

**Reproducibility:**

4: Could mostly reproduce the results, but there may be some variation because of sample variance or minor variations in their interpretation of the protocol or method.

**Reviewer Confidence:**

4: Quite sure. I tried to check the important points carefully. It's unlikely, though conceivable, that I missed something that should affect my ratings.

---

> ### Author Rebuttal · Authors · 2023-08-29
>
> We are pleased to read that the reviewer agrees regarding the effectiveness of the proposed approach and appreciates the systematicity and overall soundness of the analysis and paper.
>
> **Question: Are the generated data points expressing only a single triplet or multiple triplets as suggested by Figure 2?**\
> The triplet set sampling procedure starts by sampling the number of triplets for a given data point from a Poisson distribution. We then iteratively sample the triplets following the process outlined in the Section 3.2., which ensures coherent triplet sets until we reach the desired number of data points. In conclusion, we sample triplet *sets* and Figure 2 is consistent with the process. However, there was a typo in the caption, suggesting that the generated text is conditioned on a *triplet* rather than a *triplet set*, which we assume is the original source of confusion. The caption was updated; apologies for the inconvenience this typo has caused.
>
> **Weakness 2: More detailed analysis of the performance of SynthIE models on REBEL.**\
> We actually performed such an analysis, as reported in the first subsection of the Discussion.\
> In a nutshell, the performance drop is caused by flaws in the *text* associated with REBEL’s data points. More concretely, the text is not appropriate for *exhaustive* closed information (cIE). The GenIE models were trained on REBEL text and annotations, so they learn the systematic anomalies of the data – which, in fact, results in poor performance (14% macro-F1) of the baseline on higher quality data – but the SynthIE models, which were optimized to perform exhaustive cIE, are thrown off by them. This observation surfaces an important implication of the exhaustiveness assumption made by cIE more broadly (we discuss this in greater detail in the limitations section). In the discussion section, we outline a solution to this issue: applying the proposed synthetic data generation pipeline with different assumptions about the output distribution. We would like to note that this important implication has remained undetected thus far precisely because exhaustive annotation was out of reach before this work.\
> According to the human evaluation, despite the data anomalies, the same-sized SynthIE model is on par with GenIE in terms of macro scores (the large imbalances make the micro scores on REBEL misleading), while the predictions generated by the larger SynthIE model are of superior quality even compared to the gold annotations associated with the REBEL dataset on which GenIE was trained.
>
> **Weakness 1: The task, data, and baselines are limited, so the audience may be narrow.**\
> We would like to emphasize that, as illustrated by Figure 1, the paper proposes a general strategy for synthetic data generation and comprehensively evaluates it on the task of cIE as a purposefully chosen *case study* (a central IE task for which non-synthetic data collection is prohibitively expensive and which cannot be reliably solved by today’s LLMs directly). Overall, the proposed methodology and findings are valuable and apply to any IE or parsing task. Furthermore, any task with an inverse formulation that can be addressed more effectively by LLMs would benefit from the proposed approach as well. For more details on the last two points, see the second subsection in the Discussion.

---

### Meta-Review · Area_Chair_RKDV · 2023-09-20

**Recommendation:** 4

**Metareview:**

This work shows that LLMs can be used to synthesize data even when tasks cannot be solved directly by LLMs, particularly for problems with structured outputs. In this case, it is possible to prompt an LLM to perform the task in the reverse direction, by generating plausible input text for a target output structure. Leveraging this asymmetry in the task difficulty makes it possible to produce large-scale, high-quality data for complex tasks.  This leads to improved capabilities of smaller LLMs at structured prediction tasks, particularly IE. This is a simple but useful idea - and it has been properly investigated in the paper. The reviewers seem to like the paper as well.

---

### Decision · Program_Chairs · 2023-10-07

**Decision:**

Accept-Main

**Comment:**

This work shows that LLMs can be used to synthesize data even when tasks cannot be solved directly by LLMs, particularly for problems with structured outputs. In this case, it is possible to prompt an LLM to perform the task in the reverse direction, by generating plausible input text for a target output structure. Leveraging this asymmetry in the task difficulty makes it possible to produce large-scale, high-quality data for complex tasks.  This leads to improved capabilities of smaller LLMs at structured prediction tasks, particularly IE. This is a simple but useful idea - and it has been properly investigated in the paper. The reviewers seem to like the paper as well.